# EMPOWERED trial: protocol for a randomised control trial of digitally supported, highly personalised and measurement-based care to improve functional outcomes in young people with mood disorders

Ian B Hickie,[1] Frank Iorfino ,[1] Cathrin Rohleder,[1] Yun Ju Christine Song,[1] Alissa Nichles,[1] Natalia Zmicerevska,[1] William Capon,[1] Adam J Guastella,[1] F Markus Leweke,[1,2] Jan Scott,[3] Patrick McGorry,[4,5] Cathrine Mihalopoulos,[6] Eoin Killackey,[4,5] Min K Chong,[1] Sarah McKenna,[1] Melissa Aji,[1] Carla Gorban,[1] Jacob J Crouse,[1] Dagmar Koethe,[1] Robert Battisti,[7] Blake Hamilton,[1,8] Alice Lo,[7] Maree L Hackett,[9] Daniel F Hermens,[10] Mind Plasticity Consortia, headspace Camperdown Consortia, Elizabeth M Scott[1]

For numbered affiliations see end of article.

**Correspondence to**
Dr Frank Iorfino;
frank.iorfino@sydney.edu.au

## ABSTRACT

**Objectives** Many adolescents and young adults with emerging mood disorders do not achieve substantial improvements in education, employment, or social function after receiving standard youth mental health care. We have developed a new model of care referred to as 'highly personalised and measurement-based care' (HP&MBC). HP&MBC involves repeated assessment of multidimensional domains of morbidity to enable continuous and personalised clinical decision-making. Although measurement-based care is common in medical disease management, it is not a standard practice in mental health. This clinical effectiveness trial tests whether HP&MBC, supported by continuous digital feedback, delivers better functional improvements than standard care and digital support.

**Method and analysis** This controlled implementation trial is a PROBE study (Prospective, Randomised, Open, Blinded End-point) that comprises a multisite 24-month, assessor-blinded, follow-up study of 1500 individuals aged 15–25 years who present for mental health treatment. Eligible participants will be individually randomised (1:1) to 12 months of HP&MBC or standardised clinical care. The primary outcome measure is social and occupational functioning 12 months after trial entry, assessed by the Social and Occupational Functioning Assessment Scale. Clinical and social outcomes for all participants will be monitored for a further 12 months after cessation of active care.

**Ethics and dissemination** This clinical trial has been reviewed and approved by the Human Research Ethics Committee of the Sydney Local Health District (HREC Approval Number: X22-0042 & 2022/ETH00725, Protocol ID: BMC-YMH-003-2018, protocol version: V.3, 03/08/2022). Research findings will be disseminated through peer-reviewed journals, presentations at scientific conferences, and to user and advocacy groups. Participant data will be deidentified.

**Trial registration number** ACTRN12622000882729.

## STRENGTHS AND LIMITATIONS OF THIS STUDY

⇒ To our knowledge, this is the first large-scale effectiveness trial that tests whether early intervention and secondary prevention deliver substantive improvements in functional outcomes for young people with major mood disorders.

⇒ The trial sample will be large, and the use of minimal eligibility criteria maximises the generalisability of these findings to other youth mental health settings.

⇒ The trial introduces new service roles (i.e., 'digital navigator', 'clinical facilitator') to help clinicians and clients access the optimal package of interventions.

⇒ Standard care packages are delivered in the same setting and by the same health professional as the intervention group. So cross-over effects may attenuate between groups differences.

⇒ The availability and access to specific interventions needed to deliver enhanced care to those in the intervention group may be variable across different trial sites.

## INTRODUCTION

There has been increasing recognition of the premature death and persistent disability attributable to the major mental disorders.[1 2] The largest proportion of this excessive morbidity is attributable to mood

disorders, reflecting their early age of onset, high population prevalence, chronicity, and comorbidity.[3] While significant investments have been made in youth mental health services internationally, there is a lack of substantive evidence for which models of care are optimal for improving illness outcomes.

Mood disorders place young people at risk of prolonged socioeconomic difficulties, even when their mental ill-health subsides.[4 5] Our work has identified that up to two-thirds of young people in youth mental health services experience poor longer-term functional outcomes.[6 7] Current evidence suggests that youth mental health services primarily benefit those in the earlier stages of illness and that while brief psychological interventions are effective for reducing psychological distress, they only marginally improve functioning.[8] Further, current models of care do not appear to be well suited to those with comorbidities, mixed or attenuated symptomatology, or social and occupational complexity. Most treatment plans are focused narrowly on limited treatment choices or 'steps' for discrete disorders. They are based on average population effects or clinical experience,[9–17] and are often inaccurate and inconsistent.

The differentiation of young people with 'reasonable/good' from 'impaired/poor' functioning at presentation is a key factor to be considered (alongside other clinical variables) to determine the need for highly personalised care with the appropriate type, intensity, sequence, and duration of multidisciplinary interventions. This approach aligns with optimal models of mental health care and should be a key component of youth mental health service provision.[18] Yet, the evidence-base for health service models that guide personalised interventions for young people with mood disorders is sparse.[19–22] Furthermore, it is not standard practice to use measurement-based care (MBC) for the monitoring of symptoms and functioning to drive continuous and personalised clinical decision-making.[23–27] Highly personalised and MBC, which entails routine assessment of multidimensional outcomes and regular monitoring of an individual's response to treatment, is a core component of the chronic care model and supports better-informed clinical decisions.[28–34] These decisions may include the adjustment of treatment type and intensity. Despite good evidence for its effectiveness and its customary use in physical disease management,[31 32] it remains largely absent from youth mental health care.[9 35]

## Objectives of the study

The primary objective of this large-scale clinical effectiveness trial is to assess the effectiveness of 12 months of intensive, personally tailored, assertive care (the digitally supported highly personalised and measurement-based care (HP&MBC) package), compared with digitally supported standard clinical care. We will test whether the HP&MBC package results in a greater improvement of social and occupational functioning compared with standard clinical care. The secondary objective is to assess the mental health status of all participants 12 months after the HP&MBC and standard care interventions.

We hypothesise that while the standard care packages will be an improved offering (through greater standardisation of assessment and access to digital technology), the HP&MBC treatment packages will be superior by implementing continuous and proactive monitoring and care coordination using digital technologies and providing extensive feedback to the clinical service, the treating clinician, the young person, and their family or carer.[36]

## METHODS AND ANALYSIS
### Study design and setting
This large-scale, prospective study aims to enrol 1500 mental health treatment-seeking young people with mood disorders. The trial was designed with the aid of young people with lived experience of mental illness and is a PROBE study (Prospective, Randomised, Open, Blinded End-point). It comprises a 24 month (12 months active treatment, 12 months additional follow-up), multisite, two-arm (HP&MBC package vs standardised clinical care), randomised (1:1), blinded outcome assessor, controlled implementation trial. The trial will be conducted at the Brain and Mind Centre (The University of Sydney, Australia) and affiliated youth centres that focus on treating young people with mental illnesses. As noted below, prior to commencing the randomised controlled trial (RCT), there will be a pretrial phase (figure 1).

### Pretrial phase
The study includes a pretrial phase to allow the digital technology platform to be introduced to the clinical teams and integrated into the service procedures. This period will be used to work through any implementation issues prior to commencing the RCT. Also, it permits collection of pretrial data from each clinical service, including an audit of outcomes in routine clinical practice (e.g., rates of improvement or deterioration in social and occupational function in non-trial clinical cohorts).

### RCT and follow-up phase (~24 months)
After the pretrial phase, the RCT phase of the study commenced in early 2023, with the first participant enrolled on 1 March 2023. Participation in the trial will be 24 months following enrolment (baseline assessment), including 12 months of active treatment and 12 months additional follow-up. Five independent assessments will be conducted: at baseline, 3 months, 6 months, 12 months, and 24 months. We anticipate recruitment and randomisation of 100% of the target sample size by the end of 2025, and we estimate data collection to be completed by late 2027.

### Patient and public involvement
Young people with lived experience of mental illness were invited to participate in the study design through consultations with the Brain and Mind Centre Lived Experience

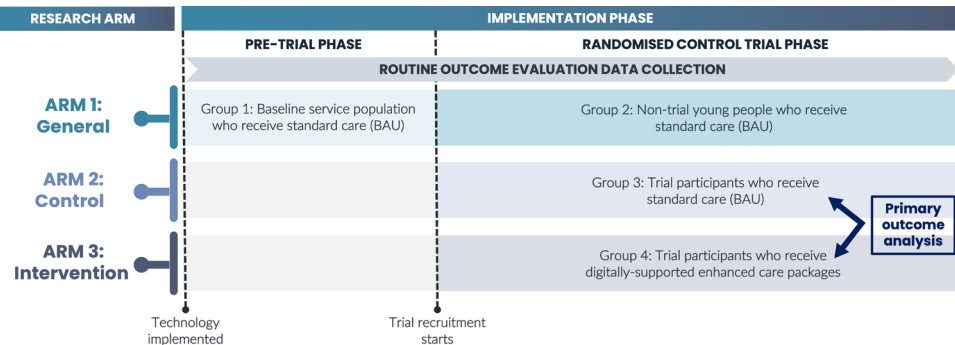

**Figure 1** Study design and service subgroups. An overview of how the trial design gives rise to distinct groups within a single participating service. There are two implementation phases and three research arms associated with this trial which result in four distinct groups for each service based on a young person's exposure and trial participation status. Group 1 is used to establish baseline outcome statistics for the service prior to the trial commencing. Groups 2, 3, and 4 differ based on the trial status, which will determine what treatments they receive. The primary outcome analysis for the randomised control trial will be between groups 3 and 4. Routine outcome evaluation data collection is ongoing from the first phase of the trial, whereby all groups will be followed up using the same processes and practices. BAU, business as usual.

Working Group. The working group consists of culturally and linguistically diverse young people aged 16–30 years. The principles underpinning the trial, and the name for this trial 'EMPOWERED', were identified by young people with lived experience (box 1).

### Study population

The study focuses on young people seeking help for psychological distress and presenting with early stage mood syndromes, characterised not only by the mix of anxiety or depressive symptoms and their impact on

<div>

**Box 1    The EMPOWERED trial principles**

1. **E**ducate: to educate young people, and their families and carers, on the potential usefulness of technology, and how routine monitoring can give them a greater say in their care journey.
2. **M**easurement-based: to improve continuous and real-time measurement of young people's symptoms and functioning, and longer-term outcomes, so that they can receive more effective care.
3. **P**ersonalised: ensuring that treatment is personalised, so that the complexity of young people's needs are recognised, documented, acted on and preserved over the care journey.
4. **O**penness: improving open communication between young people, their families and carers, and clinicians by making everyone more informed about progress in care.
5. **W**ork collaboratively: helping clinicians and young people to work collaboratively to create and respond to treatment goals by facilitating treatment monitoring, emphasising functional recovery and allowing young people to focus on assessment domains that matter most to them.
6. **E**ngage: increasing young people's engagement in care planning, by putting information about their mental health into their own hands.
7. **R**ecovery: earlier recovery through improved clinical and functional assessment, and actively monitoring social, education and employment outcomes, to ensure that young people receive earlier and more personalised care.
8. **E**nhanced **D**igitally: leverage the advanced capabilities of digital technologies to facilitate the assessment, monitoring and management of mental health problems, and support shared and informed decision-making.

</div>

functioning, but also according to stage of illness criteria: stage 1a, non-specific anxiety and depressive syndromes; stage 1b, attenuated syndromes; or stage 2, first full-threshold, major, and discrete mood syndromes.[18] Recruitment is also based on the presentation to care and existing functional impairment. This approach is consistent with the National Institute of Mental Health recommendations for conducting more integrative clinical research.[14] Approximately 10 000 individuals aged 15–25 years present to the Brain and Mind Centre and affiliated youth centres per year. We expect that 3000 individuals will meet the inclusion criteria and that about 50% of the eligible individuals will consent. In total, 1500 young people will be included (750 allocated to the active 12-month care package and 750 allocated to the standardised care package).

### Inclusion and exclusion criteria

Participation in this study will be offered to adolescents and young adults aged 15–25 years seeking help for psychological distress and classified as clinical stage 1a, 1b, or 2. The participants must have an initial Social and Occupational Functioning Assessment Scale (SOFAS) score of ≤70,[37] indicating an impaired social and occupational functioning. Additional exclusion criteria includes acute suicidal or aggressive behaviour requiring alternative care or a depressive syndrome secondary to a primary medical condition. Young people who have a clinically evident intellectual disability (IQ<70 as per medical history review) will be excluded due to the likely difficulty in completing the assessments.

### Study course and procedures

The clinical trial comprises 12 months active treatment and 12 months follow-up phase. That is, each subject will be followed for 2 years, whereby five blinded independent assessment visits will take place (figure 2).

Individuals referred to the trial will be contacted by a research team member who will provide information about the study and conduct a preliminary assessment

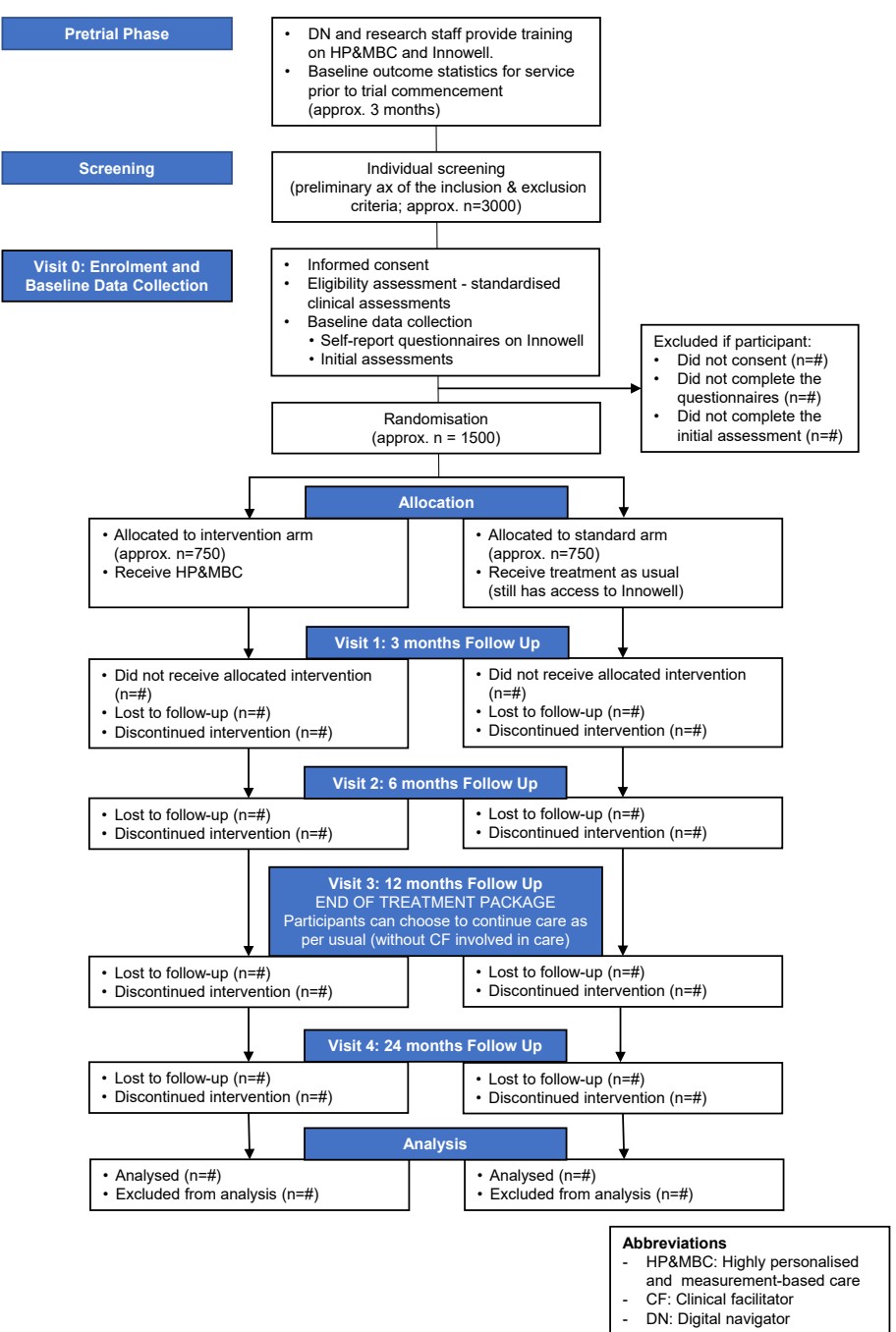

**Figure 2** Study flow diagram (consort style).

of the inclusion and exclusion criteria. Potential participants interested in taking part in the study will then be provided with a copy of the participant information statement (online supplemental material 1), and an appointment will be scheduled for an enrolment visit. During the enrolment/baseline visit (Visit 0), the study will be explained in lay terms and any questions will be answered. Following informed consent, participants will be given relevant assessments to confirm that they meet the inclusion criteria.

Participants who meet the inclusion criteria will be randomised to one of the two treatment arms using a 1:1 individual person randomisation algorithm (using Research Electronic Data Capture (REDCap))[38] taking age, gender, and treatment centre as stratification factors into account. The care packages will be delivered within the first 12 months of the study by clinicians operating within each service. During this study phase, three study visits will take place: (1) Visit 1 (3 months after trial entry); (2) Visit 2 (6 months after trial entry); and (3) Visit 3 (12 months after trial entry). A follow-up visit (Visit 4) will be conducted 12 months after completion of the care packages (figure 2). All follow-up assessments will be carried out by blinded-independent assessors from the research team.

## Care packages

To coordinate ongoing clinical care and functional recovery, real-time data feedback will be provided in both treatment arms using health information technology. As demonstrated in our longitudinal studies,[7 8 39] those with attenuated syndromes often receive only brief interventions (six or fewer sessions of psychological support) and exit services with residual high levels of impairment. There is a fundamental mismatch between the time course of impairment (typically well-established by the time the young person presents to clinical care) and the brief clinical interventions provided by early intervention services. Therefore, we now use personalised technologies to tailor, plan, and track the relationships between clinical care delivery (and sequencing) and functional recovery strategies. The real-time data feedback will support optimal combinations of indirect and direct intervention strategies until the young person achieves: (1) syndromal remission and risk reduction and (2) social and occupational recovery. This real-time data feedback will be supported by the Innowell Platform.[40]

The digitally supported HP&MBC package represents an intensive, personalised and assertive treatment package. It builds on the usual processes provided by the services at each centre, including systematic assessment and allocation of clinical care within multidisciplinary team environments. The HP&MBC package uses two key streams, namely (1) the therapeutic power of active and continuous feedback with regards to illness type, course, response to interventions, and social and economic impact of care; and (2) the capacity of new assessment and monitoring techniques to tailor treatment options—with the standardisation of those stepped-care options into an ongoing and proactive shared care plan.

The HP&MBC package includes:

1. Initial digital assessment covering the domains of symptoms, social and occupational functioning, self-harm and suicidal thoughts and behaviours (STBs), physical health, and alcohol and other substance misuse.
2. Feedback of the initial 'dashboard' of results to the user of care and family members, clinical services, and the principal treating clinician (figure 3).
3. Continuous outcome monitoring and feedback—regular review of 'dashboard' to the user of care and family members, clinical services, and principal treating clinician (monthly for first 12 months, may vary based on individual needs).
4. More detailed online, clinical, neuropsychological, and lab-based testing as recommended by digital or clinical protocols, including the use of specific individual monitoring devices (e.g., wearable activity monitors, mood monitors) to inform broad diagnostic categorisation and then assign a more specific series of highly personalised treatment options.
5. Determination of indicative subtype of depressive syndrome by incorporation of clinical factors and life course, to link to specific intervention strategies.
6. Utilisation of online shared care planning by the user of care and family members, clinical services, and principal treating clinician.

Active and continuous feedback will guide evidence-based decision-making related to treatment plans as it supports the choice of optimal combinations of interventions. The measurement-based feedback will help detect unmet needs, increase the likelihood that clinicians identify young people who are non-responsive to treatment and facilitate the process to adjust the care plan to improve the young person's outcomes.

The standard care package builds on the usual service systems (largely Medicare-funded psychological care), including systematic assessment and allocation of clinical care within multidisciplinary team environments.

The standardised care package includes:

1. Initial digital assessment covering the domains of symptoms, social and occupational function, self-harm and STBs, physical health, and alcohol and other substance misuse.
2. Feedback of the initial 'dashboard' of results to the user and treating clinician at baseline.
3. Provision of standard multidisciplinary care options and ongoing access to other relevant psychological and pharmacological options.

In this study, the following targeted therapies (over and above standard psychological care), which have been shown in various studies to have beneficial effects,[18 41–49] are of particular relevance.

1. *Social and Functional Recovery Therapies:* interventions that target social recovery include direct support to return to work, re-engaging in education or training, and social skills training to reduce isolation and improve relationships with peers and family. Key components of these interventions include setting meaningful recovery goals, establishing the external resources to support recovery, and using outreach graded behavioural experiments to re-establish functioning.
2. *Circadian Interventions:* pharmacological (e.g., agomelatine, brexpiprazole), physical (e.g., light therapy), or behavioural interventions (e.g., sleep–wake rescheduling) that target dysregulation of sleep–wake behaviours and biological circadian rhythms.
3. *Cognitive-Behaviour Therapies (CBT) and Social Therapies Groups:* CBT teaches the individual to link their feelings, thoughts and patterns of behaviours to reduce psychological distress. A greater focus on social cognition training may be needed for those with social cognitive impairment.
4. *Dialectic Behaviour Therapy (DBT):* DBT is a modified version of CBT designed to treat symptoms often associated with emotional dysregulation and poor distress tolerance such as self-harm, suicidal behaviour, and substance use. The emphasis is on moving away from harmful coping behaviours and incorporates mindfulness, distress tolerance, emotional regulation, and interpersonal effectiveness strategies.

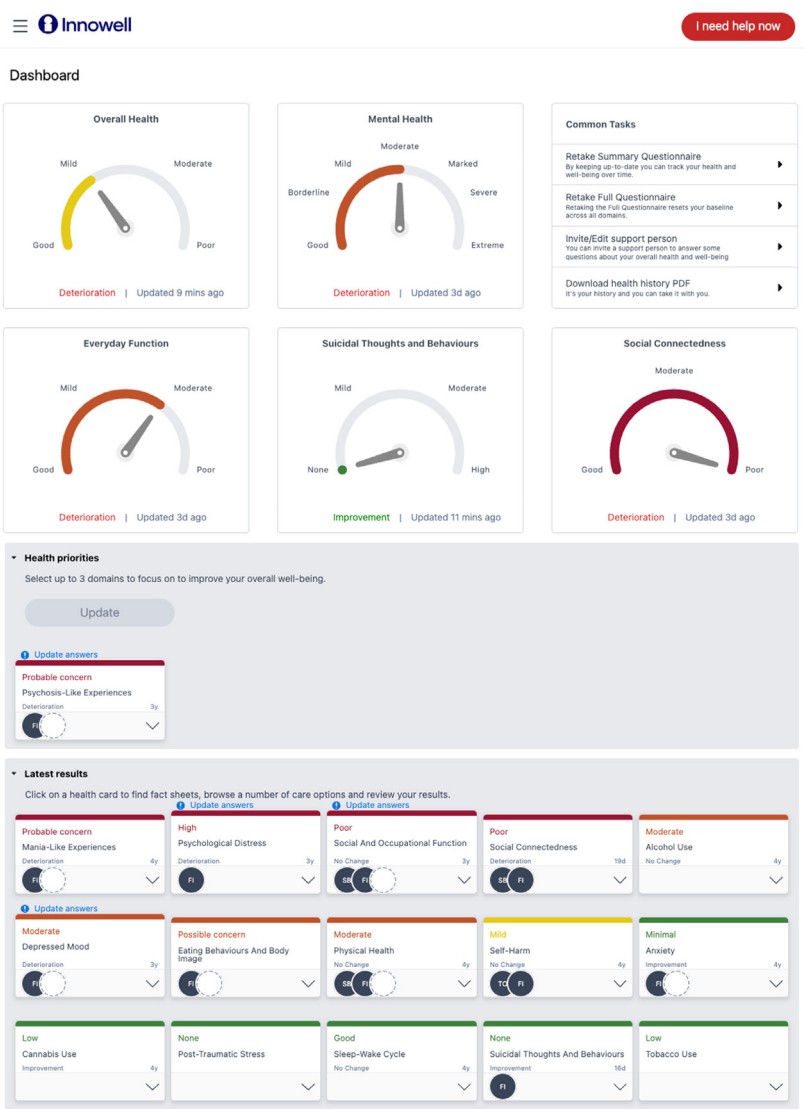

**Figure 3** An example dashboard of results from the Innowell Platform.

5. *Healthy lifestyle and cardiometabolic health targeted treatments:* the World Health Organisation (WHO) guidelines recommend that lifestyle behavioural interventions should be considered the first-line treatments for managing physical health (including cardiometabolic health) for those with severe mental illness. Psychoeducational interventions focusing on healthy lifestyle habits including diet, physical activity, and sleep practices have been shown to ameliorate both the physical and mental health concerns of young people with psychiatric disorders.

While these therapies will be available to participants in both treatment arms, those in the HP&MBC treatment group will be actively referred to the specific optimal treatment programme/programmes based on the outcomes of the continuous assessment data that will be made available to the participant and their treating clinician. In addition to the targeted therapies mentioned above, additional relevant therapies may be introduced over the course of the study as the clinical needs of participants become apparent.

## Service roles

Two additional service roles will be employed for the trial (table 1). The first is a 'clinical facilitator' who is an independent clinician focused on ensuring optimal uptake of the HP&MBC by treating clinicians. This will be achieved by working collaboratively with clinicians with the aim of reducing the additional tasks associated with enhanced and rapid communication, and tracking, interpreting, and actioning feedback. The second role is a 'digital navigator' who will operate for participants across both arms of the trial. The primary focus of the role is to provide peer support for young people to motivate them to provide outcome data; regularly remind them of the purpose of collecting data and how it can improve their treatment journey; help the young person, carer, and clinician address technical issues; and provide guidance about useful e-tools to be used in treatment. Data in the form of research observations and field notes will be collected to document the process of how these roles function

**Table 1** Role descriptions of the facilitator team

| Description of roles | Tasks | Examples |
|---|---|---|
| *Clinical facilitator*<br>*Only available for participants in the intervention arm*<br>The purpose of this role is to facilitate the use of the HP&MBC by clinicians. This is achieved by working collaboratively with clinicians with the aim of reducing burden associated with communication, tracking, and interpreting and actioning feedback. The main responsibilities of this role include:<br>1. Assisting clinicians to review and aid identification of any domains of concern (eg, increased risk or decreased social support).<br>2. Providing logistical support in making referrals for clients.<br>This role does not have any clinical responsibility towards clients as this is a support role. | ▶ Ensure that each participant has a clear functional recovery plan that has been discussed with the young person and their clinician (i.e., what is the plan and who is involved?)<br>▶ Promote and assist with the use of routine client feedback to inform personalised treatment options<br>▶ Reduce time burden for clinicians by monitoring client progress using technology and alerting clinicians if significant deteriorations/risk arise<br>▶ Performing administrative tasks to facilitate referrals, and identify appropriate treatment options recommended by the youth model<br>▶ Regularly assess with clinicians how client feedback has been used in sessions to inform treatment<br>▶ Develop a good understanding of referral options in the relevant area including community organisations, schools, public health services, online services, and apps<br>▶ Assist with identifying appropriate care options and help with the logistics of the organisation of clinical care | **SOFAS deterioration**<br>CF notes that patient X's SOFAS has deteriorated 10 points since their last report 1 month ago. CF communicates with X's psychiatrist, using their preferred communication method, letting them know that there has been a deterioration. Psychiatrist notifies CF that they have commenced a new course of treatment at their last appointment 2 weeks ago and will continue monitoring their symptoms. CF also communicates with psychologist to let them know about deterioration and notes that psychiatrist has changed medication recently. Psychologist notes that client X has recently begun exposure exercises in their weekly therapy sessions that they are finding highly distressing.<br>One month later, the client reports further deterioration to SOFAS and that they have experienced an increase in passive suicidal ideation. CF communicates this to the psychiatrist and psychologist. Psychiatrist requests DBT and CF facilitates meeting between psychologist and psychiatrist to discuss options.<br>CF also contacts three local community and public health services that offer DBT programmes and finds that Cremorne Health Centre has a spot available for client X. CF passes this information to psychiatrist to make referral. |
| *Digital navigator*<br>*Available to participants in both arms of the trial*<br>The primary focus of the role is to:<br>1. Provide peer support for clients to motivate them to provide outcome data—regularly remind the clients of the purpose of collecting data and how it can improve their treatment journey.<br>2. Help client, carer, and clinician to address other technical issues.<br>3. Provide guidance about useful e-tools (online resources and apps) to be used in treatment. | ▶ Troubleshoot any issues related to technology for clients, caregivers and clinicians<br>▶ Remind clients to complete Innowell questionnaires. Routinely follow-up with clients, through their preferred method (e.g., text, email, or face to face) to ensure regular data collection<br>▶ Research evidence-based e-tools that clinicians can confidently use as part of treatment | **Enrolment of a new participant**<br>Client X newly joined the trial. The DN will organise a brief meeting with the client to introduce Innowell and to educate them on the purpose of using the platform and its potential benefits.<br>After 1 month, DN follows up with client X to collect feedback about their experience of Innowell and whether regular reporting about their symptoms has been used by clinicians to inform treatment. Client X states that they liked how their functional scores were discussed during the session but wished that their physical health status was addressed. DN relays the feedback and suggests an app that can monitor client physical status to CF. CF alerts clinicians about client X's physical scores and promotes active response. |

CBT, cognitive-behaviour therapy; CF, clinical facilitator; DBT, dialectic behaviour therapy; DN, digital navigator; HP&MBC, highly personalised and measurement-based care; SOFAS, Social and Occupational Functioning Assessment Scale.

## Assessments

A series of standardised clinical assessments will be conducted at the enrolment visit (Visit 0) to assess inclusion and exclusion criteria (table 2), including:

1. Structured Clinical Interview for Diagnostic and Statistical Manual of Mental Disorders, Fifth Edition,[50] to assess the presence of mental health and substance use disorders.

2. A framework for clinical staging[18 51] will be applied to assess illness severity and differentiate those in the earliest phases with non-specific clinical presentations (stages 1a 'seeking help') from those at greater-risk with more specific, subthreshold presentations (stage 1b 'attenuated syndromes') and those who have reached a threshold for a progressive or recurrent disorder meeting diagnostic criteria (stage 2, 3 or 4).

**Table 2**  Overview of research assessments

| Domain | Assessment | Administration | Time points (months) | | | | |
|---|---|---|---|---|---|---|---|
| | | | 0 | 3 | 6 | 12 | 24 |
| Clinical diagnosis | Structured Clinical Interview to assess for DSM-5 Mental Health and Substance Use Disorders | Researcher administered | ✓ | | | ✓ | |
| Acute suicidal and aggressive behaviour (exclusion criteria) | CAARMS (7.3 and 5.4) | Researcher administered | ✓ | | | | |
| Social and occupational functioning | SOFAS | Researcher administered | ✓ | ✓ | ✓ | ✓ | ✓ |
| Social and occupational functioning | PSP | Researcher administered | ✓ | ✓ | ✓ | ✓ | ✓ |
| Depressive symptoms | QIDS-C | Researcher administered | ✓ | ✓ | ✓ | ✓ | ✓ |
| Illness severity | Clinical staging | Researcher administered | ✓ | ✓ | ✓ | ✓ | ✓ |
| Quality of life | ReQoL-10 | Self-report | ✓ | ✓ | ✓ | ✓ | ✓ |
| Self-harm/suicidal thoughts and behaviours | SIDAS/adaptation of the C-SSRS/B-NSS-AT | Self-report | ✓ | ✓ | ✓ | ✓ | ✓ |
| Alcohol and substance use | AUDIT-C: Alcohol use WHO-ASSIST: Alcohol and other substance use | Self-report | ✓ | ✓ | ✓ | ✓ | ✓ |
| Physical health | Height/weight/waist/BMI | Self-report | ✓ | ✓ | ✓ | ✓ | ✓ |
| Physical health | IPAQ (physical activity) | Self-report | ✓ | ✓ | ✓ | ✓ | ✓ |
| Physical health | Metabolic, inflammatory and standard clinical bloods | Researcher administered | ✓ | | ✓ | ✓ | |
| Resource use | Resource Use Questionnaire | Self-report | ✓ | ✓ | ✓ | ✓ | ✓ |

AUDIT-C, Alcohol Use Disorders Identification Test – Consumption; BMI, Body Mass Index; B-NSSI-AT, Brief Non-suicidal Self-injury Assessment Tool; CAARMS (7.3), Comprehensive Assessment of At-Risk Mental States – item 7.3; C-SSRS, Columbia-Suicide Severity Rating Scale; DSM-5, Diagnostic and Statistical Manual of Mental Disorders, fifth edition; IPAQ, International Physical Activity Questionnaire – short version; PSP, Personal and Social Performance Scale; QIDS-C, Quick Inventory of Depressive Symptomatology – Clinician-rated; ReQoL-10, Recovering Quality of Life Questionnaire (10-item version); SIDAS, Suicidal Ideation Attributes Scale; SOFAS, Social and Occupational Functioning Assessment Scale; WHO-ASSIST, WHO Alcohol, Smoking and Substance Involvement Screening Test (version 3.1).

3. SOFAS to record the clinician's judgement of overall social and occupational function.
4. A mental risk assessment to assess acute suicidal behaviour.

As summarised in table 2, individuals who fulfil all inclusion and exclusion criteria will undergo additional clinician/researcher-administered baseline assessments evaluating depressive symptomatology, personal social performance and self-report questionnaires which will be provided to collect information regarding demographics, mental and physical health history, quality of life, self-harm, STBs, alcohol and substance use, and physical health. Furthermore, blood samples will be collected to assess metabolic, inflammatory, and standard blood markers.

While social and occupational functioning, illness severity, and depressive symptoms will be assessed at every subsequent visit (Visits 1–4), the structured clinical interview will only be repeated at the end of the active treatment phase (12 months after trial entry, Visit 3; and 24 months after trial entry, Visit 4). Self-report questionnaires will be provided at each visit during the active treatment phase (Visits 1–3). Blood samples will be collected at baseline, 6 months, and 12 months after trial entry (i.e., at visits 0, 2, and 3) to monitor changes in metabolic, inflammatory, and standard blood markers. Blood test results for young people in the intervention arm will be immediately relayed to treating clinicians, to provide more data to determine the appropriate treatment for the participant. For example, non-specific immunosuppressive therapies or innovative immune therapies could be the optimal treatment approach for young people with atypical major mood or psychotic disorders.

Resource use that will also be used to estimate costs will be measured using two main procedures:

1. Participants will be asked for access to administrative datasets including the Medicare Benefits Schedule and Pharmaceutical Benefits Schedule data for the duration of the study.
2. The resource use questionnaire, used in multiple mental health economic evaluations, captures the broad range of health and welfare services used by participants and is complementary to any administrative data also included in the evaluation.[52 53]

Microcosting techniques will be used to assess the costs of the intervention. Standardised economic evaluation techniques, including incremental analysis of mean differences using generalised linear models and bootstrapping to determine confidence intervals, will also be conducted. Lifetime and population cost-effectiveness will be also determined using economic modelling techniques.

## Primary and secondary outcomes

Primary efficacy endpoint:

► Changes in social and occupational function from baseline to 12 months, as assessed by the SOFAS.
  Key secondary endpoints:
► Change from baseline in self-harm, STBs (Brief Non-suicidal Self-injury Assessment Tool, B-NSSI-AT;[54] Suicidal Ideation Attributes Scale, SIDAS;[55] C-SSRS, Columbia-Suicide Severity Rating Scale).[56]
► Change from baseline in depressive symptoms (Quick Inventory of Depressive Symptomatology, QIDS).[57]
► Change from baseline in quality of life (Recovering Quality of Life Questionnaire, ReQoL).[58 59]
► Change from baseline in alcohol and substance use (WHO Alcohol, Smoking and Substance Involvement Screening Test, WHO-ASSIST;[60] AUDIT-C, Alcohol Use Disorders Identification Test – Consumption).[61]
► Change from baseline in physical health (International Physical Activity Questionnaire – short version, IPAQ[62 63]; height, weight, waist).
► Change from baseline in metabolic, inflammatory and standard blood measures (metabolic and inflammatory markers, e.g., assessment of triglycerides, cholesterol, glucose, iron).
► Resource use as well as lifetime and population cost-effectiveness.
► Costs of the treatment packages based on detailed economic evaluation.

## Sample size calculation

This trial seeks to recruit 1500 young people, with 750 allocated to active 12-month intervention and 750 to standard clinical care. We anticipate an attrition rate of approximately 10%–20% over short-term follow-up (first 12 months) and up to 30% over the longer-term follow-up (at 24 months). Therefore, we would expect 1350 participants at 6 months follow-up (675 in each arm), 1200 participants at 12 months follow-up (600 in each arm) and 1050 participants at 2 years follow-up (525 in each arm). Assuming that we have at least 434 young people at the 2-year follow-up time point for the primary outcome analysis only, and conservatively assuming a small effect size difference of 0.2 in favour of those young people receiving the active intervention, with α=0.05, we have 95% power. For categorical secondary analyses with a small effect size of 0.2, α=0.05, and power=95%, the sample size at 2-year follow-up is 325 participants. There are also embedded subgroups for secondary analyses (e.g., by baseline suicidal behaviours, depressive subtype, alcohol or other substance misuse, and baseline SOFAS bands). For these subgroups, assuming that we have at least 195 young people at the 2-year follow-up time point for the primary outcome analysis only, and conservatively assuming a medium effect size difference of 0.3 in favour of those young people receiving the active intervention, with α=0.05, we have 95% power. For categorical secondary analyses with a medium effect size of 0.3, α=0.05, and power=95%, the sample size at 2-year follow-up is 144 participants.

## Data analysis plan

The primary outcome will be analysed using a repeated-measure linear mixed model including all available SOFAS scores measured at months 0, 3, 6, 12, and 24. Fixed effects will include the randomised group, visit as a categorical variable, and the interaction between group and visit. The baseline SOFAS score will be included as a covariate alongside sex, age, and site (stratification variables). To account for correlations between repeated measures, a random patient intercept will be included. In case of convergence issues with the inclusion of the random effect, we will replace the random effect with a repeated effect assuming a compound symmetry covariance structure. This model will be used to derive the effect of the intervention at 12 months, expressed as the adjusted mean difference and its 95% confidence interval. The effect of the intervention at other timepoints will be estimated using a similar approach.

Secondary outcomes will be analysed using a similar approach. For binary outcomes, logistic regression (binomial distribution with logit link) will be used in place of linear regression. The effect of the intervention will be estimated as the odds ratio and 95% confidence intervals and converted to an absolute risk difference using the Hummel and Wiseman method.[64] Given that linear mixed models use all data available and make valid inference under the assumption that data are missing at random, the primary analysis will not impute missing data; however, sensitivity analyses will be conducted to assess the robustness of the results under different assumptions about the missing data mechanism. A detailed statistical analysis plan including mock tables will be developed prior to unblinding and database lock.

The economic evaluation of the HP&MBC package is critical to translate this research into practice. It will comprise both a 'within-trial' design whereby the individual level costs and outcomes of the two groups

(HP&MBC and Standard Care packages) will be included in the evaluation over the duration of the trial. A modelled evaluation will be undertaken to capture full costs and consequences of HP&MBC using the results of this trial and the broader epidemiological literature to estimate likely longer-term health gains, cost impacts, and scale-up costs at the population level. The calculation of quality-adjusted life years will be conducted, thus enabling a cost-utility analysis to be undertaken. Cost-utility analyses are useful to decision-makers as they are associated with inherent value for money connotations. Detailed costing of the HP&MBC approach along with how it has been implemented within each site will be undertaken using information from the researchers, clinical staff, and budgetary personnel.

## Data management and security

All data collected for the purposes of the study will be linked to unique study ID codes and will not contain identifiable information. Data collection will be conducted only by authorised members of study staff, to whom this duty has been allocated and who are named on the Human Ethics application and Governance approvals for the trial. Research data will be stored in REDCap and electronic data generated by participant outcomes will be electronically stored via the Innowell Platform.

Any publications or reports based on this study will include only pooled results from participants. Routine internal audits of data files will ensure completeness of data collection. Data for which hardcopies are generated will be stored in both original hard copy and electronic form. Hardcopies will be retained so that comparison between electronic and original data is possible to ensure accuracy of data entry and resolve issues concerning spurious data in the electronic file. This data will be kept under (1) lock and key at trial site or (2) electronic file that is password protected and accessible only by research staff responsible for data entry or monitoring.

Monitoring will be done by investigator Yun Ju Christine Song as she is removed from the day-to-day activities and has clinical research associate experience. This will be at site initiation, after the first 50 patients are enrolled and then 6 monthly after, and at the close-out visit. The monitoring visits will involve a self-audit checklist, 10% source data verification, review of adverse events and serious adverse events, inclusion and exclusion criteria review, and a protocol deviation review.

## Participant safety

Safety reporting is subject to the National Health and Medical Research Council's guidance on Safety Monitoring and Reporting in non-therapeutic good trials. Participants do not give up any legal rights to compensation by participating in this study. If a participant suffers any injuries or complications as a result of the research project, they will be advised to contact the study team and will be assisted with arranging appropriate medical treatment.

## ETHICS AND DISSEMINATION

The study will be performed according to the Declaration of Helsinki (2008) and the International Conference on Harmonisation—Good Clinical Practice (ICH-GCP) and has been reviewed and approved by the Human Research Ethics Committee (HREC) of the Sydney Local Health District (HREC Approval Number: X22-0042 & 2022/ETH00725, Protocol ID: BMC-YMH-003-2018, protocol version: V.3, 03/08/2022). The study has been registered in the Australian New Zealand Clinical Trial Registry (ACTRN12622000882729). Any amendments will be submitted to the HREC for review prior to implementation as per HREC guidelines.

The results of this study will be disseminated as widely as possible into the scientific and broader community, including publication in peer-reviewed journals, scholarly book chapters, presentation at conferences and publication in conference proceedings. This will include one paper investigating the primary outcome measure of this study (SOFAS scores), one paper determining the economic feasibility of the HP&MBC package, and a series of papers investigating secondary outcomes (e.g., depression, suicidality). For each paper, all authors will satisfy the Vancouver criteria for authorship.

**Author affiliations**
[1]Brain and Mind Centre, The University of Sydney, Camperdown, New South Wales, Australia
[2]Faculty of Medicine Mannheim, Psychiatry and Psychotherapy, Central Institute of Mental Health, Heidelberg University, Mannheim, Germany
[3]Newcastle University, Newcastle upon Tyne, UK
[4]Centre for Youth Mental Health, University of Melbourne Australia, Parkville, Victoria, Australia
[5]Orygen, The National Centre of Excellence in Youth Mental Health, Parkville, Victoria, Australia
[6]School of Public Health and Preventive Medicine, Monash University, Clayton, Victoria, Australia
[7]Mind Plasticity, Sydney, New South Wales, Australia
[8]headspace Camperdown, Camperdown, New South Wales, Australia
[9]George Institute for Global Health, Newtown, New South Wales, Australia
[10]Thompson Institute, University of the Sunshine Coast, Birtinya, Queensland, Australia

**Acknowledgements** We are deeply indebted to all clinicians and health professionals at the participating centres who contribute to this clinical trial. Particularly to the Mind Plasticity Consortia and the headspace Camperdown Consortia.

**Contributors** IBH conceived the research idea and is the principal investigator. FI, AN, YJCS and NZ contributed to the study design and conception. CR, AN, YJCS and NZ wrote the study protocol with input from IBH, FI, WC, AJG, FML, JS, PM, CM, EK, MKC, SM, MA, CG, JC, DK, RB, BH, AL, MLH, DFH, MPC, HCC and EMS. FI and CR wrote the manuscript with input from IBH, AN, NZ, YJCS, NZ, WC, AJG, FML, JS, PM, CM, EK, MKC, SM, MA, CG, JC, DK, RB, BH, AL, MLH, DFH and EMS.

**Funding** This trial is an investigator-initiated trial funded by the NHMRC – 2020 Clinical Trials and Cohort Studies, Application ID: 2001568. Ian B Hickie is supported by a NHMRC L3 Investigator Grant (GNT2016346). Frank Iorfino is supported by a NHMRC EL1 Investigator Grant (GNT2018157). Jacob J Crouse is supported by a NHMRC Emerging Leadership Fellowship (GNT2008197).

**Competing interests** Professor Ian Hickie is the Co-Director, Health and Policy at the Brain and Mind Centre (BMC) University of Sydney, Australia. The BMC operates an early-intervention youth services at Camperdown under contract to headspace. Professor Hickie has previously led community-based and pharmaceutical industry-

supported (Wyeth, Eli Lily, Servier, Pfizer, AstraZeneca, Janssen Cilag) projects focused on the identification and better management of anxiety and depression. He is the Chief Scientific Advisor to, and a 3.2% equity shareholder in, InnoWell Pty Ltd which aims to transform mental health services through the use of innovative technologies. A/Prof Elizabeth Scott is a Research Affiliate, The University of Sydney and Consultant Psychiatrist. She has received honoraria for educational seminars related to the clinical management of depressive disorders supported by Servier and Eli-Lilly pharmaceuticals. She has participated in a national advisory board for the antidepressant compound Pristiq, manufactured by Pfizer. She was the National Coordinator of an antidepressant trial sponsored by Servier. Professor Jan Scott is a visiting professor at Diderot University, the Norwegian University of Science and Technology, Swinburne University of Technology and The University of Sydney and a 'Science without Borders' fellow (Brazil). She has received grant funding from the UK Medical Research Council and from the UK Research for Patient Benefit programme; she declares no financial or other conflict of interests in relation to the topics addressed in this paper.

**Patient and public involvement**  Patients and/or the public were involved in the design, or conduct, or reporting or dissemination plans of this research. Refer to the Methods section for further details.

**Patient consent for publication**  Not applicable.

**Provenance and peer review**  Not commissioned; externally peer reviewed.

**ORCID iDs**
Frank Iorfino http://orcid.org/0000-0003-1109-0972
Alissa Nichles http://orcid.org/0000-0001-6404-7199
Natalia Zmicerevska http://orcid.org/0000-0001-7649-4711
William Capon http://orcid.org/0000-0001-6500-9629
Adam J Guastella http://orcid.org/0000-0001-8178-4625
Jan Scott http://orcid.org/0000-0002-7203-8601
Min K Chong http://orcid.org/0000-0003-4562-731X
Sarah McKenna http://orcid.org/0000-0002-8560-6399
Jacob J Crouse http://orcid.org/0000-0002-3805-2936
Maree L Hackett http://orcid.org/0000-0003-1211-9087

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
