## [Reviewer comments · BMJ Open]

ARTICLE DETAILS

TITLE (PROVISIONAL)	The EMPOWERED trial: Protocol for a randomised control trial of digitally supported, highly personalised and measurement-based care to improve functional outcomes in young people with mood disorders
AUTHORS	Hickie, Ian; Iorfino, Frank; Rohleder, Cathrin; Song, Yun; Nichles, Alissa; Zmicerevska, Natalia; Capon, William; Guastella, Adam; Leweke, F. Markus; Scott, Jan; McGorry, Patrick; Mihalopoulos, Cathrine; Killackey, Eoin; Chong, Min Kyung; McKenna, Sarah; Aji, Melissa; Gorban, Carla; Crouse, Jacob; Koethe, Dagmar; Battisti, Robert; Hamilton, Blake; Lo, Alice; Hackett, Maree; Hermens, Daniel; Consortia, Mind Plasticity; Consortia, headspace Camperdown; Scott, Elizabeth

VERSION 1 – REVIEW

REVIEWER	Ole Haavet University of Oslo, Department of General Practice
REVIEW RETURNED	15-Mar-2023

GENERAL COMMENTS	To the authors, the protocol sets up an important study. In the youth population, I believe that it is a natural development based on current knowledge to investigate the effect of personalized and measurement-based care, and that the outcome goals are not only in health, but also social and occupational functioning. The protocol seems well thought out and well planned, but there are some aspects that I believe are missing. I am missing a timetable for the implementation of the project with the most important milestones. It would also be a strength for the protocol and the implementation of the project with an overview of planned articles, distribution of responsibilities as first and last author, and possibly also which others (project participants) participate as authors in each article (and that this happens according to the Vancouver criteria for authorship). It is interesting that immunological diseases are included in the study, but this now appears as an appendix to the main study. I am missing a better justification for this part of the protocol and a more concrete description of the relevant diseases and blood tests. Finally: The protocol uses many acronyms. It will facilitate the reading of the protocol if an explanatory overview of all the acronyms is made in a separate box.
--

REVIEWER	Nicholas Allan The Ohio State University, Psychiatry and Behavioral Health
REVIEW RETURNED	18-May-2023

GENERAL COMMENTS	Thank you for the opportunity to review this study protocol. This study presents a new model of mental health care for young adults with mood disorders called 'highly personalized and measurement-based care' (HP&MBC), which involves ongoing, detailed assessments for personalized treatment decisions. The trial tests if this approach, combined with constant digital feedback, can provide better improvements in education, employment, and social function compared to the standard youth mental health care and digital support. I had only a few suggestions for clarity. Methods and Design. It would be helpful to define what is meant by a PROBE design. Introduction Measurement-based care is introduced early but it is unclear how exactly this is operationalized.
---

VERSION 1 – AUTHOR RESPONSE

Reviewer: 1

Prof. Ole Haavet, University of Oslo, University of Oslo

Comments to the Author:

To the authors,

the protocol sets up an important study. In the youth population, I believe that it is a natural development based on current knowledge to investigate the effect of personalized and measurement-based care, and that the outcome goals are not only in health, but also social and occupational functioning. The protocol seems well thought out and well planned, but there are some aspects that I believe are missing.

I am missing a timetable for the implementation of the project with the most important milestones. We thank the reviewer for their positive comments about the manuscript. We agree that it is important to acknowledge key milestones for this study. We have included two sentences within the 'RCT and follow up phase (~24 months)' subsection that state estimates for when the trial phase will commence, when recruitment will reach 100% of our target, and when data collection will be completed: "After the pre-trial phase, the RCT phase of the study commenced in early 2023, with the first participant enrolled on 1/03/2023."... "We anticipate recruitment and randomization of 100% of the target sample size by the end of 2025, and we estimate data collection to be completed by late 2027." These estimates are based on the milestones below, which we provided to the Australian New Zealand Clinical Trials Registry (ANZCTR):

- Date of enrolment for first participant (anticipated): 15/01/23 (note: the actual first enrolment date was 1/03/23)
- Recruitment and randomization of 100% target sample size: 1/08/25
- Date completed of data collection: 1/08/27

Based on these estimates, we can infer some further details about the timeline of this project:

- Recruitment and randomization of 25% target sample size: 15/09/23
- Recruitment and randomization of 50% target sample size: 15/05/24
- Recruitment and randomization of 75% target sample size: 15/01/25
- Date completed final study report: 1/11/27 (estimate)
- Publication of results: December 2027.

It would also be a strength for the protocol and the implementation of the project with an overview of planned articles, distribution of responsibilities as first and last author, and possibly also which others (project participants) participate as authors in each article (and that this happens according to the Vancouver criteria for authorship).

We thank the reviewer for their suggestion. We have included sentences under the subheading 'Ethics and dissemination' to address this comment. This paragraph states the potential for several papers to be published using the data collected in this study and directly follows the subsection 'Data

analysis plan”. “The results of this study will be disseminated as widely as possible into the scientific and broader community, including publication in peer-reviewed journals, scholarly book chapters, presentation at conferences and publication in conference proceedings. This will include one paper investigating the primary outcome measure of this study (SOFAS scores), one paper determining the economic feasibility of the HP&MBC package, and a series of papers investigating secondary outcomes (e.g. depression, suicidality). For each paper, all authors will satisfy the Vancouver criteria for authorship.”

Importantly, these analyses will not be conducted for many years, which makes it difficult to reliably indicate which project participants will participate as authors in each article. We will ensure that all authors will satisfy the Vancouver criteria for authorship. We have included this statement in the final sentence of the ‘Ethics and dissemination’ paragraph.

It is interesting that immunological diseases are included in the study, but this now appears as an appendix to the main study. I am missing a better justification for this part of the protocol and a more concrete description of the relevant diseases and blood tests.

We agree it is important to clarify this component of the manuscript. In the ‘Assessments’ subsection, we have included more detail to explain the need for blood tests to inform measurement-based care.

“Blood test results for young people in the intervention arm will be immediately relayed to treating clinicians, to provide more data to determine the appropriate treatment for the participant. For example, non-specific immunosuppressive therapies or innovative immune therapies could be the optimal treatment approach for young people with atypical major mood or psychotic disorders.”

We have also included examples of what blood measures are being assessed in the ‘Primary and secondary outcomes’ subsection:

- “Change from baseline in metabolic, inflammatory and standard blood measures (metabolic and inflammatory markers, e.g. assessment of triglycerides, cholesterol, glucose, iron)”

The protocol uses many acronyms. It will facilitate the reading of the protocol if an explanatory overview of all the acronyms is made in a separate box.

We acknowledge the reviewer’s comment and agree that there are many acronyms used in this paper, particularly because of the large list of scales used for assessment. We thank the reviewer for suggesting an overview of all acronyms in a separate box. Below is an alphabetically ordered table, which we hope the editors can include as a separate box if they proceed to publish our manuscript.

Glossary of abbreviations

Abbreviation Term

AUDIT-C Alcohol Use Disorders Identification Test – Consumption;

BMI Body Mass Index;

B-NSSI-AT Brief Non-suicidal Self-injury Assessment Tool;

CAARMS (7.3) Comprehensive Assessment of At-Risk Mental States – item 7.3;

CBT Cognitive-Behaviour Therapy

C-SSRS Columbia-Suicide Severity Rating Scale;

DSM-5 Diagnostic and Statistical Manual of Mental Disorders, 5th edition;

HP& MBC Highly Personalised and Measurement-Based Care

HREC Human Research Ethics Committee

ICH-GCP International Conference on Harmonisation - Good Clinical Practice

IPAQ International Physical Activity Questionnaire – short version;

IQ Intelligence Quotient

MBC Measurement-based Care

MBS Medicare Benefits Schedule

NHMRC National Health and Medical Research Council

PBS Pharmaceutical Benefits Schedule

PROBE Prospective, randomized, open, blinded endpoint evaluation

PSP Personal and Social Performance Scale;
QIDS-C Quick Inventory of Depressive Symptomatology – Clinician-rated;
RCT Randomised Controlled Trial
ReQoL-10 Recovering Quality of Life Questionnaire (10-item version);
SIDAS Suicidal Ideation Attributes Scale;
SOFAS Social and Occupational Functioning Assessment Scale
STBs Suicidal Thoughts and Behaviours
WHO-ASSIST World Health Organisation Alcohol, Smoking and Substance Involvement Screening Test (version 3.1).

Reviewer: 2

Dr. Nicholas Allan, The Ohio State University

Comments to the Author:

Thank you for the opportunity to review this study protocol. This study presents a new model of mental health care for young adults with mood disorders called 'highly personalized and measurement-based care' (HP&MBC), which involves ongoing, detailed assessments for personalized treatment decisions. The trial tests if this approach, combined with constant digital feedback, can provide better improvements in education, employment, and social function compared to the standard youth mental health care and digital support. I had only a few suggestions for clarity.

Methods and Design.

It would be helpful to define what is meant by a PROBE design.

We thank the reviewer for identifying the need to clarify what is meant by a PROBE design. We have included a description in brackets of this acronym in the two places it is mentioned. Firstly, within the Method and analysis paragraph of the abstract: "This controlled implementation trial is a PROBE study (Prospective Randomized Open, Blinded End-point) that comprises a multi-site 24-month, assessor-blinded, follow-up study of 1500 individuals aged 15-25 years who present for mental health treatment."

And secondly, in the Study design and setting paragraph of the Methods and analysis section of the manuscript: "The trial was designed with the aid of young people with lived experience of mental illness and is a PROBE study (Prospective Randomized Open, Blinded End-point). It comprises a 24-month (12 months active treatment, 12 months additional follow-up), multi-site, two-arm (HP&MBC care package vs standardised clinical care), randomised (1:1), blinded outcome assessor, controlled implementation trial."

Introduction

Measurement-based care is introduced early but it is unclear how exactly this is operationalized.

We agree with the reviewer that the initial version of this manuscript briefly introduced measurement-based care but lacked an explanation of what measurement-based care entails in the context of mental health treatment. We have added in additional detail within the introduction to expand on this point. "Furthermore, it is not standard practice to use measurement-based care (MBC) for the monitoring of symptoms and functioning to drive continuous and personalised clinical decision making[28-32]. Highly personalised and measurement-based care, which entails routine assessment of multidimensional outcomes and regular monitoring of an individual's response to treatment, is a core component of the chronic care model and supports better-informed clinical decisions[33-39]. These decisions may include the adjustment of treatment type and intensity. Despite good evidence for its effectiveness and its customary use in physical disease management[36, 37], it remains largely absent from youth mental health care[13, 40]."

Though more detail in the introduction could be included, we include specific detail about the 'highly personalised and measurement-based care' package within the 'Care Packages' subsection of the 'Methods and Analysis' section. "The digitally supported HP&MBC care package represents an

intensive, personalised and assertive treatment package. It builds on the usual processes provided by the services at each centre, including systematic assessment and allocation of clinical care within multidisciplinary team environments. The HP&MBC care package uses two key streams, namely (a) the therapeutic power of active and continuous feedback with regards to illness type, course, response to interventions and social and economic impact of care; and (b) the capacity of new assessment and monitoring techniques to tailor treatment options – with the standardisation of those stepped-care options into an on-going and proactive shared care plan.

The HP&MBC enhanced care package includes:

- (i) Initial digital assessment covering the domains of symptoms, social and occupational functioning, self-harm and suicidal thoughts and behaviours (STBs), physical health and alcohol and other substance misuse;
- (ii) Feedback of the initial ‘dashboard’ of results to the user of care and family members, clinical services and the principal treating clinician (Figure 3);
- (iii) Continuous outcome monitoring and feedback – Regular review of ‘dashboard’ to the user of care and family members, clinical services, and principal treating clinician (monthly for first 12-months, may vary based on individual needs);
- (iv) More detailed online, clinical, neuropsychological and lab-based testing as recommended by digital or clinical protocols, including use of specific individual monitoring devices (e.g., wearable activity monitors, mood monitors) to inform broad diagnostic categorisation and then assign a more specific series of highly personalised treatment options;
- (v) Determination of indicative sub-type of depressive syndrome by incorporation of clinical factors and life course, to link to specific intervention strategies;
- (vi) Utilisation of online shared care planning by the user of care and family members, clinical services, and principal treating clinician;

Active and continuous feedback will guide evidence-based decision making related to treatment plans as it supports the choice of optimal combinations of interventions. The measurement-based feedback will help detect unmet needs, increase the likelihood that clinicians identify young persons who are non-responsive to treatment, and facilitate the process to adjust the plan of care to improve young person outcomes.”

VERSION 2 – REVIEW

REVIEWER	Ole Haavet University of Oslo, Department of General Practice
REVIEW RETURNED	24-Jul-2023
GENERAL COMMENTS	The reviewer completed the checklist but made no further comments.